Using cluster and rest redistribution set structures as alternatives to resistance training prescription method based on velocity loss thresholds

http://orcid.org/0000-0002-0900-9410 Jukic Ivan 1 2 ivan.jukic@aut.ac.nz
Helms Eric R. 1
McGuigan Michael R. 1
http://orcid.org/0000-0003-0608-8755 García-Ramos Amador 3 4
1 Sport Performance Research Institute New Zealand (SPRINZ), Auckland University of Technology , Auckland , New Zealand
2 School of Engineering, Computer and Mathematical Sciences, Auckland University of Technology , Auckland , New Zealand
3 Department of Physical Education and Sport, Faculty of Sport Sciences, University of Granada , Granada , Spain
4 Department of Sports Sciences and Physical Conditioning, Faculty of Education, Universidad Católica de la Santísima Concepción , Concepción , Chile
Butcher Scotty
Electronic publication date: 2022 Mar 29
Publication date: 2022
Volume: 10
Electronic Location ID: e13195
Received 2021 Dec 16; Accepted 2022 Mar 9
Copyright: © 2022 Jukic et al.
Copyright year: 2022
Copyright holder: Jukic et al.
License: This is an open access article distributed under the terms of the Creative Commons Attribution License, which permits unrestricted use, distribution, reproduction and adaptation in any medium and for any purpose provided that it is properly attributed. For attribution, the original author(s), title, publication source (PeerJ) and either DOI or URL of the article must be cited.
License URL: https://creativecommons.org/licenses/by/4.0/

Keywords: Velocity-based training, Fatigue, Training prescription, Training monitoring, Strength training

Funding: The authors received no funding for this work.

==============================
Background

The purpose of this study was to compare the effects of cluster (CS), rest redistribution (RR) and traditional (TS) set configurations on acute neuromuscular performance, and to determine the viability of using CS and RR as alternatives to training prescription based on velocity loss (VL).

Methods

Thirty-one resistance-trained men performed, in a randomised order, three experimental sessions consisting of the squat (SQ) and bench press (BP) exercises performed against the 10-repetition maximum load using CS (three sets of six repetitions; 30 s of intra-set rest every two repetitions; 3 min of inter-set rest), RR (9 sets of two repetitions; 45 s of inter-set rest), and TS (3 sets of 6 repetitions; 3 min of inter-set rest), set configurations.

Results

Linear mixed-effects model analysis revealed that participants had significantly lower VL (p = 0.0005) during CS and RR than TS. Generalised mixed-effects model analysis yielded significant main effects of set structure (p < 0.0001; RR > CS > TS), exercise (p < 0.0001; SQ > BP), and set number (p = 0.0006; Set 1 > Set 2 > Set 3) for maintaining repetition velocity above a 20% VL threshold.

Conclusions

These findings suggest that CS and RR are effective at reducing the overall fatigue-included decrease in velocity compared to TS and allow the majority of repetitions to be completed with less than 20% VL. Therefore, both CS and RR can be used to manage fatigue during resistance training, and as alternatives to training prescription method based on 20% VL threshold.

Introduction

The beneficial effects of resistance training (RT) on muscular strength, power, speed, endurance, balance, coordination, and hypertrophy are widely recognised (Suchomel, Nimphius & Stone, 2016; Suchomel et al., 2018). In addition, RT is often used for injury prevention and rehabilitation purposes and has an important role in general well-being due to numerous beneficial effects on health and quality of life (O’Connor, Herring & Caravalho, 2010; Feigenbaum & Pollock, 1999). However, adaptations to RT may differ depending upon several factors such as the choice and order of the exercises, training intensity, volume, frequency, rest interval duration, set structure, and velocity of the repetitions (Schoenfeld, Ogborn & Krieger, 2017; Grgic et al., 2018a, 2018b; Pareja-Blanco et al., 2014; Jukic et al., 2020, 2021). Of these, training volume and intensity have received the majority of attention in the literature (Schoenfeld, Ogborn & Krieger, 2017; Schoenfeld et al., 2017). On that note, exercise intensity during RT has traditionally been determined as the load lifted relative to the one-repetition maximum (%1RM) while set volume has frequently been prescribed according to a theoretical maximum number of repetitions per set that can be performed against a given %1RM until reaching muscular failure (Bompa & Haff, 2009). Although this way of prescribing training loads and volume is relatively simple and practical and can be managed with relative ease with large groups of people, it does not account for physiological and psychological stressors that can affect an individual’s day-to-day performance (Mann, Ivey & Sayers, 2015). For instance, maximal strength can fluctuate daily when an individual is fatigued or significantly increase within a few weeks due to training adaptation (Padulo et al., 2012). Furthermore, the number of repetitions that can be completed against a given %1RM has been shown to be both individual- and exercise-specific (Richens & Cleather, 2014; Sánchez-Medina & González-Badillo, 2011). To combat these issues, the velocity-based training (VBT) approach to RT has been shown as a highly effective and reliable methodology for RT monitoring and prescription (González-Badillo & Sánchez-Medina, 2010; Sánchez-Medina & González-Badillo, 2011).

Among other advantages, VBT allows for the objective assessment of the neuromuscular fatigue incurred during a set by monitoring the velocity loss (VL) of the repetitions (Sánchez-Medina & González-Badillo, 2011). This is because performing multiple repetitions in a consecutive fashion without rest between repetitions (i.e., traditional sets [TS]) inevitably results in fatigue accumulation, especially when large number of repetitions are performed, and the loads used (Jukic et al., 2020). In addition, monitoring VL reached in each set serves as a precise method of quantifying the level of effort (i.e., repetitions performed with respect to the maximum number that can be completed) (Sánchez-Medina & González-Badillo, 2011). Since level of effort is an important factor in determining the acute responses (Pareja-Blanco et al., 2018) and subsequent adaptations to RT (Pareja-Blanco et al., 2017), many studies using VBT approaches to RT have examined the effects of training with different magnitudes of VL on maximal strength, hypertrophy, muscle endurance and performance of athletic tasks. In brief, higher VL (>20%) maximised hypertrophic adaptations but resulted in a significant reduction in the IIX muscle fiber phenotype (Pareja-Blanco et al., 2017, 2020) whereas lower VL (≤20%) resulted in similar improvements in muscle strength and endurance as well as performance of sport-specific tasks such as vertical jumping and sprinting (Pareja-Blanco et al., 2017). However, Pareja-Blanco et al. (2020) recently showed that here might be an upper and lower VL threshold that should be prescribed during RT to induce optimal training adaptations. It was concluded that moderate VL thresholds (i.e., VL10 and VL20) should be chosen to optimise adaptations to RT since VL0 seemed to induce levels of fatigue that were too low to maximise adaptations, whereas VL40 did not promote further muscle strength and hypertrophy compared to VL20.

Despite the benefits of monitoring repetition velocity during RT, doing so requires the use of valid and reliable velocity measuring devices (e.g., linear position transducers). While the cost of these devices has decreased in recent years, they are still not affordable to many. Even if velocity measuring devices are available, their use with groups of people can be an impractical task (e.g., team sports, group sessions). In that regard, more heuristic approaches to controlling movement velocity during RT could prove beneficial. Cluster sets (CS), where intra-set rest periods are implemented in addition to the pre-existing inter-set rest periods, are a straight-forward method to reduce fatigue while allowing higher velocity repetitions to be performed during RT (Jukic et al., 2020). However, CS extend total training time relative to TS which might not always be feasible from a practical standpoint. A more time efficient alternative to CS is to simply redistribute the total rest time of TS structures to include shorter and more frequent rest intervals. Just like CS, this set structure, known as rest redistribution (RR), has been shown to be effective—albeit to a lesser extent—in alleviating acute mechanical, metabolic, and perceptual markers of fatigue during RT (Jukic et al., 2020), and thus effective at inducing positive training adaptations (Jukic et al., 2021). With these points in mind, it is possible that both CS and RR set structures could be used as an a-priori alternative to commonly used VL thresholds. Indeed, Tufano et al. (2017) observed that 12-s inter-repetition rest periods allowed for 36 consecutive back squat repetitions to be performed without dropping below VL20 while examining the effects of CS and RR on mechanical performance whereas Jukic & Tufano (2019) recently showed that RR allowed almost all repetitions (~17.5 out of 18) in a clean pull exercise to be performed with less than 20% VL regardless of the load used across three sets. However, these studies used either CS or RR protocols in isolation, did not have a TS protocol, and used only lower body exercises in their protocols. Since both CS and RR could affect RT performance differentially (Jukic et al., 2020, 2021) with these effects often being exercise specific (Latella et al., 2019), it would be beneficial to examine both CS and RR as an alternative to VL thresholds (e.g., VL20) across multiple sets and exercises within the same study.

To shed light on whether CS and RR could be used as an a-priori alternative to VL, we examined the effects of TS, CS and RR on (a) VL reached during back squat (SQ) and bench press (BP) exercises across multiple sets; and (b) the number of instances at which VL20 was not exceeded. Based on previous research (Jukic & Tufano, 2019; Jukic et al., 2020, 2021), we hypothesised that: (a) lower VL will be experienced during CS and RR compared to TS during both SQ and BP across all sets (CS < RR < TS); and (b) Both CS and RR would allow for the greatest number of repetitions to be performed above the VL20 threshold (CS > RR > TS) and could potentially serve as an a-priori alternative to VL20 thresholds.

Materials and Methods

Participants

Thirty-one resistance-trained men volunteered to participate in this study (mean ± standard deviation [SD]: age = 21.3 ± 2.3 years (range = 18–30 years); body mass: 78.4 ± 12.9 kg; body height = 1.76 ± 0.07 m; SQ 10RM = 78.5 ± 12.7 kg; BP 10RM = 65.9 ± 13.7 kg). To be eligible for this study, participants had to be free from any musculoskeletal injury and familiarised with RT while having at least 1 year of experience in performing BP and SQ exercises with a proper technique. During the study, participants were instructed to avoid any strenuous exercise. Written informed consent was obtained after the participants received an oral explanation of the purpose and potential risks of the study. All procedures were conducted in accordance with the Helsinki Declaration and approved by the University of Granada’s Ethics Committee (IRB approval: 935/CEIH/2019).

Study design

A randomised cross-over design was used to examine the effects of different set configurations on velocity loss and to determine whether alternative set structures (i.e., CS and RR) could be used as an alternative to 20% velocity loss threshold during strength-oriented RT sessions. For this purpose, participants reported to the laboratory on four occasions within a 2-week period. During the first visit, the 10RMs during SQ and BP exercises were determined. Thereafter, participants reported to the laboratory on three occasions (i.e., experimental sessions) during which they performed BP and SQ exercises using different set structures. All experimental sessions were performed in a randomised order with 48 to 72 h of rest between sessions. Barbell velocity was collected during all repetitions. Throughout the sessions, strong verbal encouragement and visual velocity feedback were provided to ensure that participants performed the concentric phase of the exercises with the maximal intent. All sessions for the same participants were held at the same time of day (±1 h) to minimize the effects of the circadian rhythm on physical performance.

Ten-repetition maximum session: day 1

Before the commencement of the incremental loading test, participants completed a standardized warm-up consisting of running at a self-selected low intensity pace for 5 min, and dynamic stretching for 5 min. Thereafter, an incremental loading test was performed to determine the load associated with a mean velocity of 0.70 ms–1 and 0.55 ms–1 (≈10RM, 80% of 1RM) for SQ and BP, respectively. The first external load in both exercises was the unloaded barbell of the Smith machine (20 kg), and weight discs were added until reaching the mean velocities described above. Thenceforth, participants were instructed to perform a set of repetitions to failure against these loads to determine their 10RM. If it was clear that the participant was able to perform less or more than 10 repetitions with a given load, the set was immediately terminated, and 5 min of rest was provided before the next 10RM attempt with a modified load. The magnitude of the load change was decided based on the mutual consensus between the participants and the experienced researcher. The 10RM load was determined in no more than three attempts for all participants. During the SQ exercise, participants were required to squat until the top of their thighs were parallel to the floor without pausing at the bottom. In addition, participants were instructed to self-select their grip width (Pérez-Castilla et al., 2020, 2021) and perform the BP exercise such that the bar was pressed concentrically immediately after touching the chest without a pause.

Experimental sessions: days 2–4

Following the 10RM session, participants returned to the laboratory on another three separate occasions during two consecutive weeks. During these sessions, participants had to perform SQ and BP against the 10RM load using traditional (TS; 3 sets of 6 continuous repetitions with 3 min of inter-set rest), cluster (CS; 3 sets of 6 repetitions with 30 s of intra-set rest every 2 repetitions and 3 min of inter-set rest), and rest redistribution (RR; 9 sets of 2 repetitions with 45 s of inter-set rest) set structures with 10 min of rest between the SQ and BP exercises. Sessions were separated by 48 to 72 h of rest and were performed in a randomised order with exercises performed in the same order during all three sessions. Each experimental session was preceded by the standardised warm-up followed by 10 bodyweight SQ, 5 SQ with 50% of 10RM load, 2 SQ with the 10RM load, 10 push-ups, 5 BP with 50% of the 10RM load, and 2 BP with the 10RM load.

Data acquisition

All repetitions in the present study were performed in a FFittech Smith Machine (Taiwan, China) which did not have a counterweight system and with the weight of the unloaded barbell being 20 kg. Mean velocity of all repetitions was recorded using a dynamic measurement system (T-Force System; Ergotech, Murcia, Spain). This system consists of a linear velocity transducer interfaced to a computer by means of a 14-bit analog-to-digital data acquisition board and custom software. Instantaneous velocity was sampled at 1,000 Hz and smoothed using a fourth order low-pass Butterworth filter with no phase shift and 10 Hz cut-off frequency. Reliability of this system has been reported elsewhere (Pérez-Castilla et al., 2019; González-Badillo & Sánchez-Medina, 2010) and its cable was vertically attached to the right side of the barbell, between the hands and the loaded barbell sleeves, with the Velcro strap.

Statistical analysis

All raw data on velocity loss were normally distributed as determined by graphical inspection and the indicator value range for skewness and kurtosis (Trochim & Donnelly, 2006; Gravetter et al., 2020; Field, Miles & Field, 2012). Descriptive data are presented as medians and interquartile ranges unless otherwise stated. To examine the effects of different set structures on VL during each set, linear mixed effects models were used. For this purpose, velocity loss served as an outcome measure whereas set structure (three levels) was treated as a fixed effect. In addition, set number (three levels) and exercise (two levels) were also included as fixed effects and modelled as an interaction with the set structure.

The same approach as described above was also used to test whether CS and RR set structures maintained barbell velocity at less than a 20% loss. However, for this purpose, generalised mixed effects models were used. Since the number of instances at which participants stayed above 20% VL threshold was the outcome variable in this case, a binomial error distribution was specified with a logit-link function to predict the odds of staying above 20% velocity loss threshold as a function of the set structure and interactive effects of the exercise and set number (fixed effects).

For all models, participants were treated as random effects, while random slopes were also introduced in the models as long as their addition did not result in a convergence error. Therefore, due to the inclusion of both fixed and random effects, restricted maximum likelihood estimation was used for evaluation of the models. Furthermore, their contribution—and the contribution of modelled interactions among predictors—to the explanatory power of any of the explored models was examined using a likelihood ratio test, deviance statistic and Akaike information criterion score before selecting the final model in order to obtain the best-fit model while maintaining model parsimony. The final linear mixed-effects model included interaction between the set structures, exercises and set numbers as fixed effects and participants included as random effects. However, the final generalised mixed-effects model was identical except also including a random slope for set structures. For linear mixed-effects models, estimated marginal means and 95% confidence intervals were calculated and presented with comparisons made using post-hoc Holm-Bonferroni adjustments. For generalised mixed-effects models odds ratios as well as predicted probabilities were evaluated and presented to aid interpretation of the findings.

Since regression-based models can be sensitive to variables that are correlated, the variance inflation factors for all predictor parameters used in the linear mixed-effects model were inspected to check for multi-collinearity. In addition, autocorrelation diagnostics were performed to confirm the independence of the observations. For linear models, a Gaussian distribution was assumed, and the approximate normal distribution of model residuals was checked to confirm goodness of fit. To ensure the assumptions of the model were met, the plotted residuals were also checked to ensure homoscedasticity prior to utilising the results of the model. To validate the assumptions of the generalised mixed-effects model, tests for uniformity of residuals, under and over dispersion, outliers and zero-inflation were performed using simulation-based approach (which works comparably to parametric bootstrapping—see (Hartig, 2021)), which confirmed the absence of significant problems with the model fit.

All statistical analyses were conducted in R language and environment for statistical computing using the lme4, emmeans, and ggeffects packages while model assumptions were checked using the performance and DHARMa packages (4.0.5; R Core Team, Vienna, Austria).

Results

For descriptive purposes, raw data with medians and interquartile intervals as well as their distribution is presented in Fig. 1. Results from the linear mixed-effects model (interactions and main effects) are described in the text, whereas estimated marginal means with 95% confidence intervals and pairwise comparisons are illustrated in Fig. 2. The final linear mixed-effects model on the velocity loss experienced during TS, CS and RR revealed a non-significant interaction between the set structure, set number, and exercise (F = 0.793; p = 0.530). Similarly, exercise × set number and set structure × set number interactions did not reach statistical significance (F = 0.693–0.772; p = 0.500–0.544). However, set structure × exercise interaction was significant (F = 9.846; p = 0.0005). In addition, significant main effects of set structure (F = 97.811; p < 0.0001; TS > CS > RR), exercise (F = 281.153; p < 0.0001; BP > SQ) and set number (F = 7.770; p = 0.0005; Set 3 > Set 2 > Set 1) were observed for the amount of VL reached.

Figure 1 Medians and interquartile ranges (i.e., middle 50% of the data between 75th and 25th percentiles) for mean velocity loss across all set structures, exercises and training sets.

The dashed line represents a 20% velocity loss threshold whereas each dot represents one data point (3.22% of the whole data) and together with density represents the distribution of the data. BP, Bench press; CS, Cluster set structure; RR, Rest redistribution set structure; SQ, Back squat; TS, Traditional set structure.

Figure 2 Estimated marginal means with 95% confidence intervals for the mean velocity loss across all set structures, exercises, and training sets.

BP, Bench press; CS, Cluster set structure; RR, Rest redistribution set structure; SQ, Back squat; TS, Traditional set structure; **Significantly greater decline than CS and RR (p < 0.001); **#Significantly greater decline than CS (p < 0.05) and RR (p < 0.001); #Significantly greater decline than RR (p < 0.05); ##Significantly greater decline than RR (p < 0.001).

The final generalised linear mixed-effects model for the number of instances participants were able to stay above the 20% VL threshold revealed no significant set structure × exercise × set number, exercise × set number, set structure × exercise, and set structure × set number interactions (χ2 = 0.015–4.417; p = 0.959–0.110). However, significant main effects of set structure (χ2 = 45.356; p < 0.0001; RR > CS > TS), exercise (χ2 = 162.987; p < 0.0001; SQ > BP), and set number (χ2 = 14.813; p = 0.0006; Set 1 > Set 2 > Set 3) were found for staying above the 20% VL threshold. Specific comparisons (odds ratios) are presented in Table 1. Predicted probabilities for staying above the 20% VL threshold broken down by the set structure, exercise and set number are illustrated in Fig. 3.

Table 1 Pairwise comparisons between different set structures for the odds of staying above 20% velocity loss threshold broken down by exercise and training set.

Description	Statistics	
Comparison	Exercise	Set number	Odds ratio	SE	Z score	p value	
CS/RR	Bench press	1	0.500	0.181	–1.914	0.056	
CS/TS	Bench press	1	2.600	0.792	3.138	0.003	
RR/TS	Bench press	1	5.205	1.946	4.412	0.000	
CS/RR	Back squat	1	0.413	0.273	–1.339	0.180	
CS/TS	Back squat	1	3.767	1.783	2.802	0.010	
RR/TS	Back squat	1	9.126	5.625	3.587	0.001	
CS/RR	Bench press	2	0.330	0.120	–3.060	0.004	
CS/TS	Bench press	2	2.360	0.688	2.947	0.004	
RR/TS	Bench press	2	7.145	2.711	5.183	0.000	
CS/RR	Back squat	2	0.525	0.327	–1.036	0.300	
CS/TS	Back squat	2	4.948	2.285	3.462	0.001	
RR/TS	Back squat	2	9.431	5.335	3.967	0.000	
CS/RR	Bench press	3	0.589	0.187	–1.670	0.095	
CS/TS	Bench press	3	2.463	0.697	3.183	0.003	
RR/TS	Bench press	3	4.180	1.419	4.214	0.000	
CS/RR	Back squat	3	0.888	0.454	–0.233	0.816	
CS/TS	Back squat	3	4.646	1.985	3.596	0.001	
RR/TS	Back squat	3	5.234	2.420	3.581	0.001	
Note:

CS, Cluster set structure; TS, Traditional set structure; RR, Rest redistribution set structure; SE, Standard error.

Figure 3 Predicted probabilities with 95% confidence intervals for staying above 20% velocity loss threshold for all set structures, exercises, and training sets. For specific comparisons, please refer to Table 1.

BP, Bench press; CS, Cluster set structure; RR, Rest redistribution set structure; SQ, Back squat; TS, Traditional set structure.

Discussion

The purpose of this study was to examine the effects of CS, RR, and TS on neuromuscular performance and to determine the viability of using CS and RR as alternatives to prescribing VL20% thresholds across multiple BP and SQ sets. The main findings of this study revealed that (a) the inclusion of 30 s intra-set intervals during CS, and creating shorter but more frequent rest periods during RR prevented the VL experienced during TS with RR being the most effective set structure for this purpose; (b) these effects were more pronounced during BP than SQ (i.e., the magnitude of VL was lower); (c) both CS and RR allowed the vast majority of repetitions to be performed above VL20%; and (d) velocity was better maintained above VL20% during the BP compared to SQ during CS and RR, with RR also being more effective than CS for maintaining velocity during the BP exercise. Taken together, data from the current study show that CS and RR are effective at reducing the overall fatigue-induced decrease in velocity compared with TS while also allowing the vast majority of repetitions to be completed above VL20%—though the magnitude of these effects are likely to be exercise-specific.

As expected, significantly lower VL was observed during both CS and RR when compared to TS, regardless of the exercise and the set number—a finding that is in agreement with the literature (Jukic et al., 2020). However, our first hypothesis was only partially supported since, although both CS and RR were effective at reducing VL experienced during TS, RR was more effective than CS. The way in which CS and RR set structures were constructed could potentially explain the greater effects of RR over CS on VL. Namely, RR set structure consisted of 9 sets of 2 repetitions with 45 s of inter-set rest whereas CS consisted of 3 min of rest after every set of 6 repetitions which was interspersed by 30 s of rest after every 2 repetitions. Due to this setup, inter-set of RR (45 s) can be aligned and compared with the intra-set rest of CS (30 s). In this regard, longer rest periods after every 2 repetitions during RR than CS might have allowed better energy substrate restoration, while only 30 s of rest during CS caused some residual fatigue to be carried over from the previous set of 2 repetitions, subsequently resulting in greater VL compared to RR (see Supplemental Files S1 and S2 for visualisation). However, although plausible, this is speculative since energy substrate restoration was not measured in the present study. Regardless of the mechanism behind this finding, it must be noted that RR set structures are also more efficient as they do not extend total training time like CS, but still attenuate neuromuscular fatigue (and seemingly to a greater extent than CS, at least in the present study). This is particularly important for practitioners who are often time constrained. Furthermore, the magnitude of the effect of CS and RR on VL, when compared to TS, was lower for the SQ than the BP exercise since much lower VL was experienced during TS, and subsequently CS and RR, in the SQ than the BP exercise. Therefore, when considering the findings from the present study, findings from a recent meta-analysis (Latella et al., 2019), and distinct neuromuscular and perceptual fatigue profiles reported in the literature for upper- and lower-body muscles (Vernillo et al., 2018; Mayo, Iglesias-Soler & Fernández-Del-Olmo, 2014), the effects of CS and RR on the overall neuromuscular performance might be exercise specific.

Although reducing neuromuscular fatigue, as quantified by VL incurred in a training set during resistance training, is a very beneficial feature of CS and RR set structures, the VL experienced might still be unnecessarily high (i.e., higher than 25%) which could result in sub-maximal training adaptations (Pareja-Blanco et al., 2017, 2020). In this regard, a significant body of literature now suggests terminating training sets after VL20–25% has been reached. The rationale behind this suggestion is based on the findings from longitudinal studies which collectively show how VL20–25% is the most efficient VL threshold as it induces hypertrophic adaptations (Pareja-Blanco et al., 2020), allows for positive neuromuscular adaptations (Pareja-Blanco et al., 2020), preserves the expression of the fast twitch muscle fibres with considerably lower training volume being performed compared to higher VL thresholds (Pareja-Blanco et al., 2017). However, valid and reliable measuring devices are needed to prescribe sets using VL which are still not affordable to many, and therefore making this training methodology inaccessible to many. The present study shows the viability of both CS and RR to potentially serve as cost-free alternatives to VL20% resistance training prescription. Specifically, the participants were able to perform the vast majority of repetitions during both CS and RR with less than 20% VL compared to TS. Therefore, CS and RR were effective not only at allowing for higher absolute velocities during the whole training session (average velocity across sets), but also at minimising the degree of fatigue (i.e., velocity loss) experienced at the end of each training set. The present results agree with the findings from the only two other studies—to our knowledge—that have taken a similar approach to assess the ability of CS and RR to maintain velocity above certain thresholds (Tufano et al., 2017; Jukic & Tufano, 2019). However, those studies used either CS or RR protocols in isolation, did not have a TS protocol, or used only lower body exercises. Therefore, the present study expands on previous findings demonstrating that the effects of CS and RR, compared to TS, were more pronounced during the BP exercise, with RR also being more effective than CS in maintaining repetition velocity above VL20%. As stated, the greater effects of RR compared to CS could be attributed to how these set structures were constructed (i.e., longer more frequent rest periods during RR). However, the greater benefits of both CS and RR during the BP could likely be explained by the smaller muscle groups involved in this exercise—which results in more localised fatigue—compared with the SQ where the fatigue is distributed among a greater amount of muscle mass. The relative position of the sticking region in these exercises may also explain these VL differences, as the SQ allows more time/distance for force generation after such region. This probably also explains why overall VL was greater in the BP compared to SQ exercise. In turn, participants managed to maintain repetition velocity in the SQ exercise even during TS (~21%), although their velocity maintenance was still considerably higher during CS and RR (~10%). In this regard, it can be concluded that CS and RR will provide greater benefits for the BP exercise due to the greater localised fatigue that is typically experienced in this compared to the SQ exercise, but also that CS and RR will allow for the extremely high velocity maintenance in the SQ exercise. Thus, practitioners should be aware that the same VL threshold (or the use of the same CS or RR setup) may produce divergent training adaptations when used for different exercises. With that said, the use of CS and RR during resistance training can be of a great practical value as it can produce similar outcomes as prescribing VL20% (Pareja-Blanco et al., 2017, 2020) while avoiding the financial and potential logistical constraints associated with the use of velocity tracking devices.

Limitations and directions for future research

The results of this research expand on previous studies investigating the effects of CS and RR on resistance training performance, and their use as cost-free alternatives to resistance training prescription via VL thresholds. However, it should be noted that our findings do not necessarily transfer, at least not to the same extent, to other exercises, and free-weight alternatives of BP and SQ exercises since our participants performed BP and SQ exercises in the Smith machine. In addition, since our participants were only recreationally trained these findings may not be generalisable to athletic or sedentary populations. Therefore, future studies should investigate the effects of CS and RR on resistance training performance in other populations as well as other commonly used exercises during resistance training. Finally, we only looked at the effects of CS and RR on VL20% with a single load, set and repetition scheme. Future research should investigate different set, repetition and loading schemes of CS and RR and their association with different VL thresholds. Doing so will provide sport professionals who don’t have access to VBT devices additional options for mimicking the benefits of VBT in a more practical way.

Conclusions

The use of CS and RR attenuated VL experienced during TS across multiple sets of both BP and SQ exercises. In addition, the use of CS and RR allowed for the vast majority of the repetitions to be performed above VL20%. These beneficial effects of CS and RR were greater in the BP exercise likely due to the smaller muscle groups involved in this exercise, thus resulting in more localised fatigue accumulation compared to the SQ exercise. While this suggests that the magnitude of the effect might be exercise-specific, velocity maintenance in the SQ exercise is still expected to be considerably greater during CS and RR than TS. Importantly, for the purpose of maximising mechanical outputs in the shortest time period, RR emerged as the most efficient set structure among the three since it allowed for the lowest VL among the three set structures, and greatest number of repetitions to be performed above VL20% without extending the total training time—which was the case during CS. Collectively, while both CS and RR are effective tools for reducing the overall fatigue experienced during resistance training, they also allow repetition velocity to be maintained above VL20%. Therefore, the beneficial effects of terminating training sets after VL20% demonstrated in prior literature can be achieved by using CS and RR, while avoiding the financial and logistical constraints associated with the use of velocity tracking devices.

Supplemental Information

Supplemental Information 1 The average velocity of every repetition performed across three sets in the back squat exercise.

The dashed line represents a 20% velocity loss threshold whereas each dot represents average velocity of every repetition across three sets expressed as percentage of the fastest repetition. CS, Cluster set structure; RR, Rest redistribution set structure; SQ, Back squat; TS, Traditional set structure

Click here for additional data file.

Supplemental Information 2 The average velocity of every repetition performed across three sets in the bench press exercise.

The dashed line represents a 20% velocity loss threshold whereas each dot represents average velocity of every repetition across three sets expressed as percentage of the fastest repetition. BP, Bench press; CS, Cluster set structure; RR, Rest redistribution set structure; TS, Traditional set structure

Click here for additional data file.

Supplemental Information 3 Dataset for linear mixed-effects model analysis.

Click here for additional data file.

Supplemental Information 4 Dataset for generalised linear mixed-effects model analysis.

Click here for additional data file.

Additional Information and Declarations

Competing Interests

Author Contributions

Human Ethics

Ethics

Data Availability

Amador García Ramos is an Academic Editor for PeerJ.

Ivan Jukic conceived and designed the experiments, analyzed the data, prepared figures and/or tables, authored or reviewed drafts of the paper, and approved the final draft.

Eric R. Helms conceived and designed the experiments, authored or reviewed drafts of the paper, and approved the final draft.

Michael R. McGuigan conceived and designed the experiments, authored or reviewed drafts of the paper, and approved the final draft.

Amador García-Ramos conceived and designed the experiments, performed the experiments, authored or reviewed drafts of the paper, and approved the final draft.

The following information was supplied relating to ethical approvals (i.e., approving body and any reference numbers):

All procedures were conducted in accordance with the Helsinki Declaration and approved by the University of Granada’s Ethics Committee (IRB approval: 935/CEIH/2019).

The following information was supplied relating to ethical approvals (i.e., approving body and any reference numbers):

The University of Granada.

The following information was supplied regarding data availability:

The datasets required to reproduce linear-mixed effects and generalised linear-mixed effects models analyses are available in the Supplemental Files.

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
