# Peer review of "Using cluster and rest redistribution set structures as alternatives to resistance training prescription method based on velocity loss thresholds"

_PeerJ, doi:10.7717/peerj.13195_

## Round 0.1 · original submission · Major Revisions

Thank you for your contribution. This is a very interesting study and shows promise for publication. Please address, in detail, the comments from both reviewers. I look forward to your revised version.
Scotty

Reviewer 1 ·

Basic reporting

The introduction is well organized, but we note the absence of some studies that have already analyzed the possibility of monitoring velocity loss in the set when instruments to measure execution velocity are not available. For example, Gonzalez-Badillo et al. (2017) Velocity Loss as a Variable for Monitoring Resistance Exercise. Int J Sports Med 38: 217-225; Rodriguez-Rosell et al. (2020) Relationship Between Velocity Loss and Repetitions in Reserve in the Bench Press and Back Squat Exercises. J Strength Cond Res 34(9): 2537-2547.
Figures and tables are acceptable. They provide sufficient information according to the results obtained in the study.

Experimental design

According to the authors, the purpose of this study was to examine the effects of CS, RR, and TS on neuromuscular performance and to determine the viability of using CS and RR as alternatives to prescribing VL20% thresholds across multiple BP and SQ sets. However, the study does not appear to have examined "the effect on neuromuscular performance," but only the viability of using CS and RR to control VL20%.

A weak aspect of the design is the determination of the 10RM load. On the one hand, the same repetitions are not performed with the same percentage in both studied exercises (see the articles cited previously). Specifically, it is unlikely that the average number of maximum repetitions at 80% will exceed six repetitions in the squat. Naturally, these repetitions refer to a squat (full squat), not a squat to parallel. In this sense, determining the thigh parallel without intermediate stop may be unreliable. What control has been performed to accept that all repetitions were done with a similar flexion? Moreover, with this type of test, any athlete is being forced to perform one or more tests, up to three, according to the study's authors, of 10RM. The authors should indicate what type of athlete/sport and at what ages it would be appropriate to apply this type of test. With the data from the two uncited articles, it is possible to know the velocity loss in the set with any load without performing any XRM test.

Not all subjects can perform the same repetitions with the same percentage (see articles not cited in the manuscript). This circumstance means that not all subjects will have trained with the same relative intensity if all subjects have reached the 10RM load. Moreover, even if the percentage was the same for all participants, not all will lose the same velocity when doing the same repetitions (see the previous articles), so the fatigue for the same repetitions will be different. In addition, velocity losses for the same repetitions are different depending on the relative intensity at which the training is performed. Therefore, if applied in a workout, the identical rest between repetitions and sets will produce a different effect.

The authors state that “the bar was pressed concentrically immediately after touching the chest without a pause to ensure the most reliable performance.” We assume that the authors are aware that some other study concludes that with the chest stop, the results are more reliable. They should cite this study and try to explain their decision.

Validity of the findings

Lines 277-84. The conclusions that “the study findings suggest that CS and RR are effective at reducing the overall fatigue-included decrease in velocity compared to TS” does not seem relevant. All the “main findings” of the study are logical and known without the study's need. It is common knowledge that the longer the recovery time between repetitions, the less fatigue and velocity loss in the set. If the recovery times between repetitions had been somewhat longer, the velocity loss would have been less. That is, the CS and RR would have shown "greater validity."

The study provides little helpful information, as it sticks to a single load. According to the design, it could not be applied to any other load. Nevertheless, as we have indicated previously, with the data from the two uncited articles, it is possible to know the velocity loss in the set with any load.

The study is justified because losing at most 20% of the velocity in the set may have a better neuromuscular effect than losing more velocity. However, an essential question arises to be solved that may render the study useless. The effect of 20% velocity losses without rest between repetitions may differ from 20% loss with intermediate rests between repetitions or groups of sets. In the studies cited in this manuscript, which serve as the basis for its justification, there was no pause between repetitions.

Lines 327-29 “CS and RR can be used not only do decrease the overall amount of neuromuscular fatigue experienced during resistance training, but also to ensure velocity remains high during training.”
This statement does not mean that CS and RR provide two applications because a lower velocity loss is the same as experiencing less fatigue.

As can be deduced from the above, the statements made in lines 324 25 (the present study shows the viability of both CS and RR to potentially serve as cost-free alternatives to VL20% resistance training prescription) and 383-86 (the beneficial effects of terminating training sets after VL20% demonstrated in prior literature can be achieved by using CS and RR, while avoiding the financial and logistical constraints associated with the use of velocity tracking devices) cannot be applied and, therefore, do not present utility.

Additional comments

No additional comments

Reviewer 2 ·

Basic reporting

General comments:
Interesting study showing the efficacy of alternative set structures as alternative methods to velocity loss thresholds, which may certainly have utility in team-based sports or large group settings. I commend the authors for their work. To the best of my knowledge, they have performed appropriate statistical analyses with the raw data provided to investigate the purpose of their manuscript. However, I have some requested revisions as addressed below.

Specific comments:
Line 49 – 50:
This sentence may confuse the reader. I would suggest changing ‘above 20% VL’ to read ‘less than 20% VL’.

Line 87:
Please remove ‘or even greater’, as there were no significant differences in muscle strength in this study cited by Pareja-Blanco et al. (DOI: 10.1111/sms.12678).

Line 113:
This sentence may confuse the reader. I would suggest changing ‘above VL20’ to read ‘at less than VL20’.

Line 171:
I think ‘things’ is a typo. Please change to ‘things’ to say ‘thighs’ or appropriate terminology.

Line 209:
This sentence may confuse the reader. I would suggest changing ‘above the 20% threshold’ to read ‘at less than 20% VL threshold’.

Line 317:
Please change ‘VL20%’ to state ‘VL20 – 25%’, since although VL20% may be suggested in the squat from Pareja-Blanco et al. (DOI: 10.1111/sms.12678) and Pareja-Blanco et al. (DOI: 10.1249/MSS.0000000000002295), VL25% may be suggested in the bench press from Pareja-Blanco et al. (DOI: 10.1111/sms.13775), and you investigated both exercises (squat and bench press) in your study.

Line 319:
Please change ‘VL20%’ to state ‘VL20 – 25%’ for the same reason as mentioned in my previous comment.

Line 326:
This sentence may confuse the reader. I would suggest changing ‘above the VL20%’ to read ‘at less than VL20%’.

Line 342:
Please change ‘production’ to ‘generation’ for further clarity.

Line 378 – 381:
Please change the following sentence, “Importantly, RR emerged as the most efficient set structure among the three since it allowed for the lowest VL among the three set structures, and greatest number of repetitions to be performed above VL20% without extending the total training time – which was the case during CS”.
RR is not necessarily 'the most efficient’ set structure because it allowed for the lowest VL and the greatest number of repetitions to be performed at less than 20% VL, as the available evidence does not suggest that VL should be as low as possible for strength nor hypertrophy. Please highlight how both the systematic review and meta-analysis by Jukic et al. (DOI: 10.1007/s40279-020-01423-4) and Davies et al. (DOI: 10.1007/s40279-020-01408-3) found no differences in strength and no differences in hypertrophy when comparing alternative set structures to traditional sets; therefore, alternative set structures do not appear to be superior nor inferior to traditional sets for strength nor hypertrophy. With that being said, you could potentially highlight how RR may be beneficial for team-based sports that involve high velocity movements, desire velocity-oriented profiles, and greater power outputs with lower loads, but it is crucial to highlight that this does not impact strength and hypertrophy when the total repetitions and load are the same between alternative set structures and traditional sets based on the two available SRMAs. Also, Davies et al. found that on average alternative set structures required more total training time than traditional sets; therefore, RR is not necessarily 'the most efficient' based on the broader available evidence. You may want to include that integrating different velocity loss thresholds (and both alternative set structures as well as traditional sets) may be beneficial in a periodized model as both have advantages and disadvantages on the acute responses and favored chronic adaptations. With everything said, I greatly appreciated the novelty of the study, well-written manuscript, and thorough statistical analyses. Awesome work!

Experimental design

no comment

Validity of the findings

no comment

---

## Round 0.2 · accepted · Accept

I would like to thank you for your thorough responses to the reviewers, which provided appropriate context, referencing, and clarification. You attention to detail and candid rebuttals were appreciated. Well done!

Reviewer 2 ·

Basic reporting

Thank you for addressing my comments.

Experimental design

no comment

Validity of the findings

no comment

Additional comments

no comment